REPORT

# In situ cryo-electron tomography reveals filamentous actin within the microtubule lumen

Danielle M. Paul[1]*, Judith Mantell[2]*, Ufuk Borucu[3], Jennifer Coombs[2], Katherine J. Surridge[2], John M. Squire[1,4], Paul Verkade[2], and Mark P. Dodding[2]

**Microtubules and filamentous (F-) actin engage in complex interactions to drive many cellular processes from subcellular organization to cell division and migration. This is thought to be largely controlled by proteins that interface between the two structurally distinct cytoskeletal components. Here, we use cryo-electron tomography to demonstrate that the microtubule lumen can be occupied by extended segments of F-actin in small molecule–induced, microtubule-based, cellular projections. We uncover an unexpected versatility in cytoskeletal form that may prompt a significant development of our current models of cellular architecture and offer a new experimental approach for the in situ study of microtubule structure and contents.**

## Introduction

Understanding of the lumenal contents of cytoplasmic microtubules has been classically driven by EM–based analysis in cells and tissues (Burton, 1984; Peters and Vaughn, 1967; Rodríguez Echandía et al., 1968), and most recently by high resolution cryo-EM (Atherton et al., 2018; Bouchet-Marquis et al., 2007; Garvalov et al., 2006; Grange et al., 2017; Koning et al., 2008). Contents of cytoplasmic microtubules are thought to be restricted to globular proteins such as tubulin-modifying enzymes (Coombes et al., 2016), although higher order structures have been observed in sperm flagella microtubules and cilia (Ichikawa and Bui, 2018; Zabeo et al., 2018). Unequivocal identification of lumenal proteins in a cellular context remains a challenge as the dimensions of the microtubule may confound super-resolution fluorescence imaging techniques, and there are potential issues with antibody-epitope accessibility in this confined environment. Cryo-EM solves the challenge of spatial resolution, but can be limited by the technical requirement for very thin samples (Schur, 2019), so studies have mainly focused on microtubules at the cell periphery or within neuronal processes (Atherton et al., 2018; Bouchet-Marquis et al., 2007; Garvalov et al., 2006; Grange et al., 2017; Koning et al., 2008). We recently identified a compound, 3,5-dibromo-N′-{[2,5-dimethyl-1-(3-nitrophenyl)-1H-pyrrol-3-yl]methylene}-4-hydroxybenzohydrazide, named "kinesore," that targets the motor protein kinesin-1, promoting extensive remodeling of the microtubule network (Randall et al., 2017). Kinesore treatment results in dynamic looping and bundling of microtubules within the cytoplasm and their extrusion from the cell body as membrane-bound projections. This renders microtubules from within the cell accessible

for cryo-EM studies (Cross and Dodding, 2019; Randall et al., 2017). Here we describe the discovery of filamentous actin (F-actin) inside the microtubule lumen through in situ cryo-electron tomography (cryo-ET) analysis of these small molecule–induced projections.

## Results and discussion

### Formation and ultrastructure of small molecule–induced, microtubule-based projections

To begin ultrastructural analysis of projections in their native state, HAP1 cells, incubated with a fluorescent membrane stain before kinesore treatment, were prepared for cryo–correlative light EM (CLEM). This allowed the unambiguous identification of projections using fluorescence microscopy that could be correlated with images from the electron microscope (Fig. 1 A). Consistent with our earlier immunofluorescence imaging study (Randall et al., 2017), abundant projections were composed of closely aligned microtubules, and we also occasionally observed vesicular structures within swellings as the periphery. Our previous live-imaging of GFP-tubulin expressing HeLa cells suggested that the projections were initially formed through the extrusion of microtubule loops (Randall et al., 2017). Live-imaging of SiR-tubulin labeled microtubules in the HAP1 cells used here confirmed this and showed that microtubules are progressively added through additional loop extrusion events (Fig. S1 and Video 1), providing a rationale for the formation of extended microtubule bundles. Consistent with this live imaging, EM analysis of regions proximal to the cell body revealed a greater number of microtubules and distinct looped bundles (Fig. 1 B), with an intermicrotubule spacing of

[1]School of Physiology, Pharmacology and Neuroscience, Faculty of Life Sciences, University of Bristol, Bristol, United Kingdom; [2]School of Biochemistry, Faculty of Life Sciences, University of Bristol, Bristol, United Kingdom; [3]GW4 Facility for High-Resolution Electron Cryo-Microscopy, University of Bristol, Bristol, United Kingdom; [4]Department of Metabolism, Digestion and Reproduction, Imperial College, London, United Kingdom.

*D.M. Paul and J. Mantell contributed equally to this paper; Correspondence to Mark P. Dodding: mark.dodding@bristol.ac.uk; Paul Verkade: p.verkade@bristol.ac.uk.



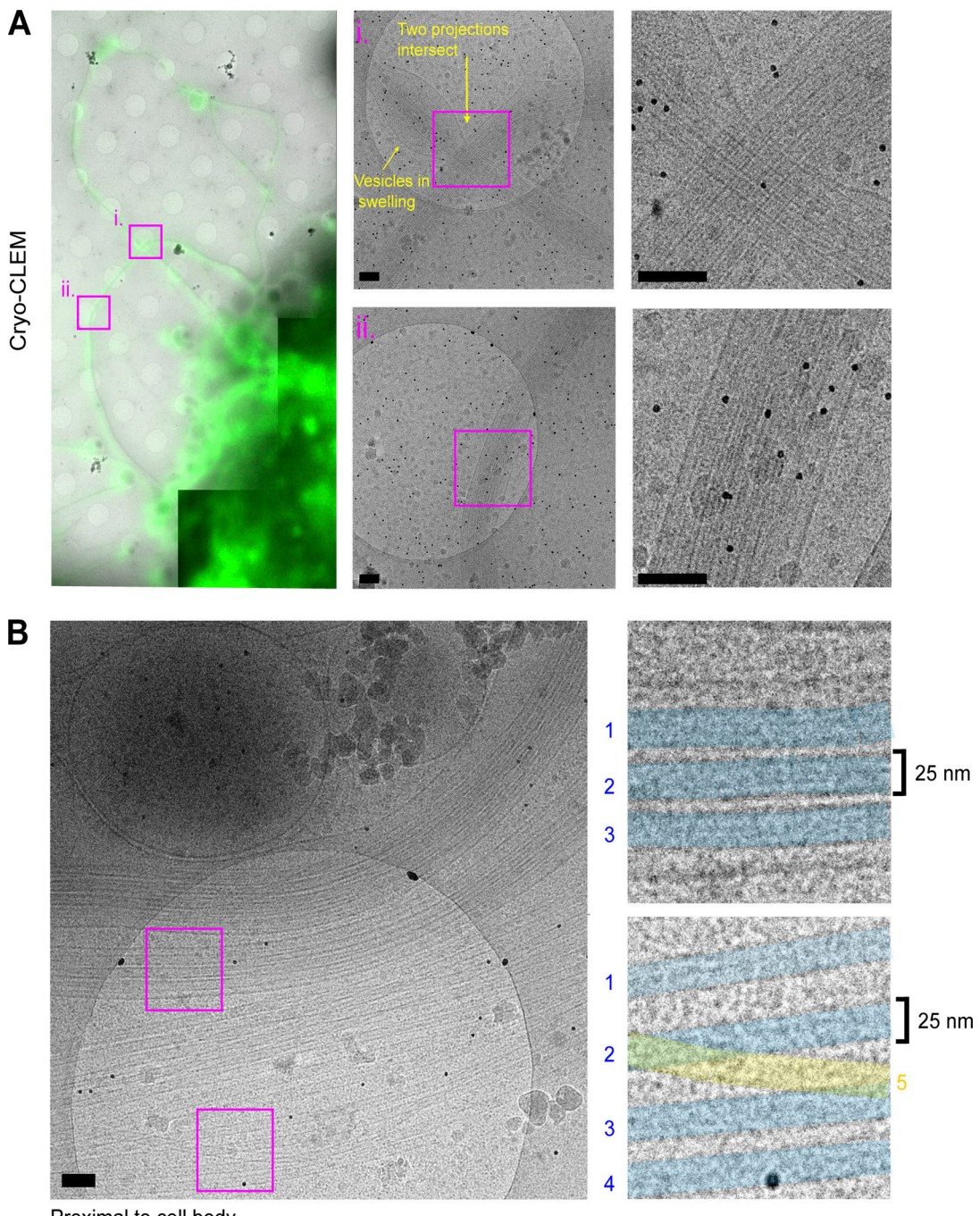

Figure 1.   **Cryo-CLEM analysis of kinesore-induced projections.** HAP1 cells grown on Quantifoil gold EM grids were stained with Cell Mask green to identify plasma membrane and treated with kinesore (100 µM) for 1 h before plunge-freezing and imaging using fluorescence and EM. **(A)** Low magnification images show fluorescence (green) and EM image (grays) overlay of cells with projections emerging. Regions boxed i and ii are shown at higher magnification and expanded on the right. Black dots are 10 nm gold fiducials. **(B)** A single EM image acquired proximal to the cell body with a large number of microtubules. Boxes show microtubules with similar alignments (shaded blue, 1–3 [top] and 1–4 [bottom]) and an example of a single microtubule that crosses a bundle of aligned microtubules (yellow, 5 [bottom]). Scale bars are 100 nm.

10–25 nm (blue shading), although some were also observed to traverse bundles (yellow shading). These observations are similar to in vitro EM studies of kinesin-mediated microtubule–microtubule cross-linking (≤25 nm spacing; Andrews et al., 1993) and measurements of the distance kinesin-1 holds its cargoes from the microtubule surface (≈17 nm; Kerssemakers et al., 2006).

## Cryo-ET reveals actin-like filaments with the lumens of extruded microtubules

Satisfied that we could confidently identify projections in the electron microscope, additional samples were analyzed without the fluorescence imaging step (avoiding ice contamination) by 3D reconstruction of tomography tilt series (Fig. 2). Transverse

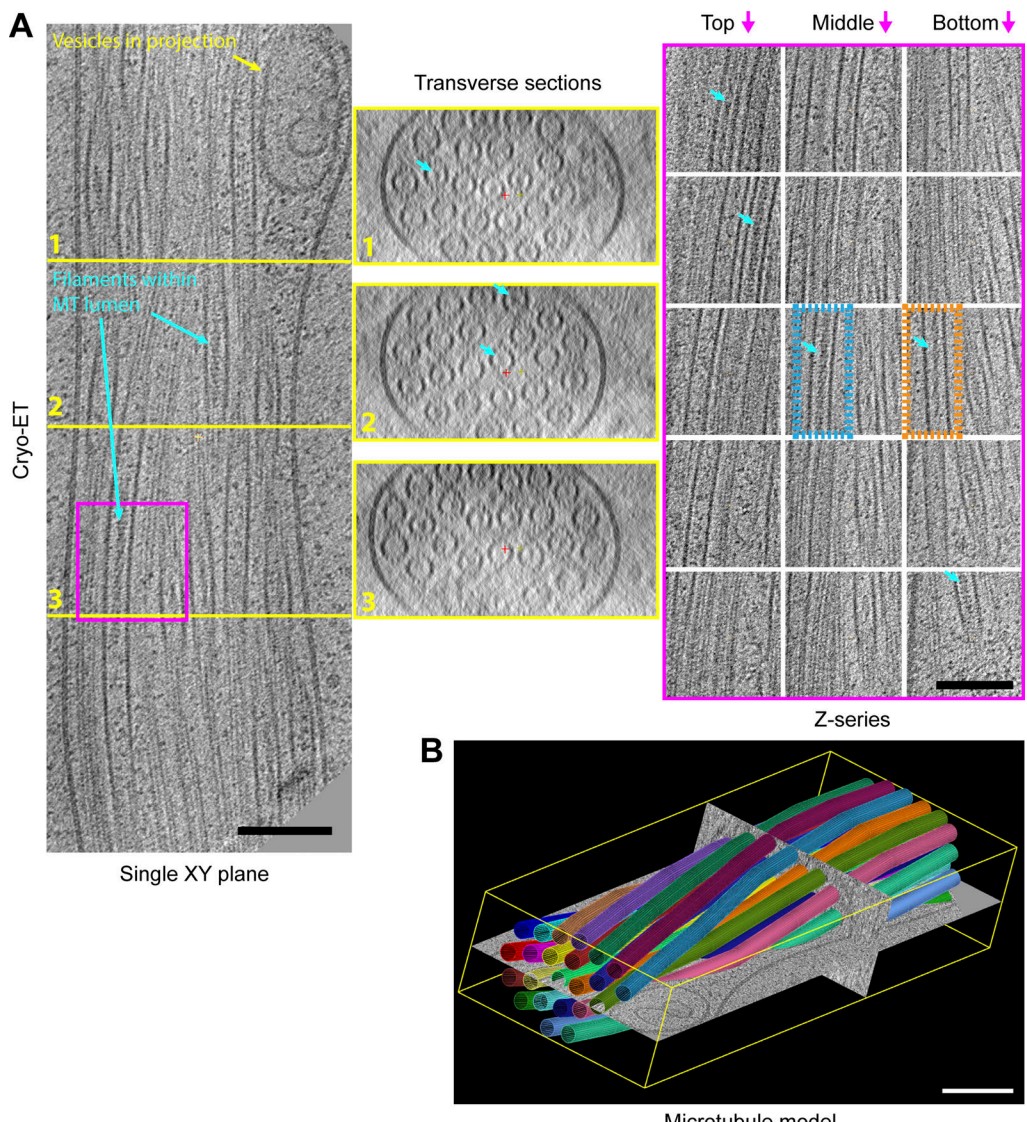

Figure 2. **Cryo-ET reveals actin-like filaments within the microtubule lumen. (A)** An XY plane (left), a series of transverse sections (middle), and a Z-series (right) though a 3D reconstruction of 23-microtubule (MT) projection. Sections that bisect the microtubule lumen reveal a mixture of particles and filamentous material with apparent helical properties. Light blue arrows highlight several lumenal filaments. The orange box region highlights filaments of the Class I form and the blue box the Class II, as described in more detail in Fig. 4. **(B)** A 3D model where microtubules are represented as colored 24 nm diameter cylinder segments (corresponds to Video 2). Microtubules are organized in a closely packed twisted membrane-bound bundle with some vesicular structures in swellings at the periphery. Scale bars are 100 nm.

sections through projections revealed that they could contain variable numbers of microtubules (ranging from 4 to >30; Fig. 2 and Fig. S2). A section of a 23-microtubule projection is shown in detail. Microtubules are organized in a twisted bundle, and vesicles are excluded in swellings proximal to the limiting membrane. Both transverse sections and serial images of Z-sections show that the microtubules are intact (Video 2). Surprisingly, within the microtubule lumens, as well as globular structures, we could clearly observe extended filamentous density with helical character (blue arrows). Two examples of such filaments are boxed in orange and blue on the Z-section panels in Fig. 2 and are also visible as density within the microtubule lumen of transverse sections (see also Fig. S2). The size and helical appearance of these lumenal filaments are consistent with those of F-actin (5–9 nm; Chou and Pollard, 2019; Fujii et al., 2010) which, at least in principle, could reside within the ≈15–16 nm diameter lumenal space (Nogales et al., 1999).

To assess whether kinesore-induced projections do indeed contain actin, methanol-fixed cells (to optimally preserve microtubules) were stained with antibodies against actin and tubulin. This revealed patches and puncta of actin along microtubules within the projections (Fig. 3 A). Actin antibodies may not discriminate between G- and F-actin, so equivalent samples were prepared using PFA fixation and phalloidin staining for F-actin. Under these fixation conditions, projections were less well preserved, with fragmentation in β-tubulin staining, but actin patches were more prominent and appeared to bridge gaps in tubulin staining (Fig. 3 B). This raises the

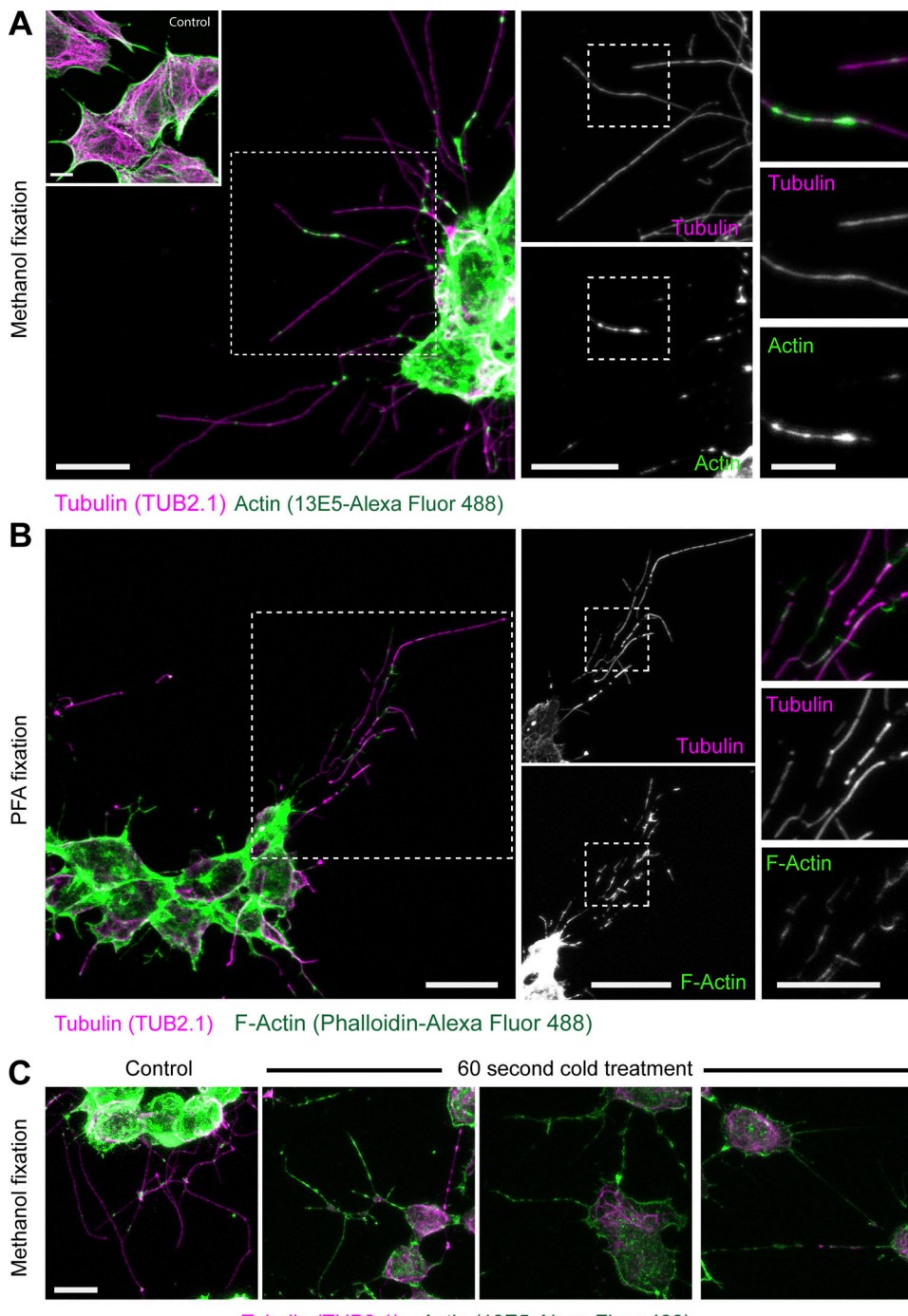

Figure 3. **Kinesore-induced projections contain both tubulin and F-actin.** HAP1 cells were treated with kinesore (100 µM) for 1 h and fixed with either ice-cold methanol (A) or 4% PFA (B). **(A)** Methanol-fixed cells were stained with antibodies against tubulin (TUB 2.1, detected by anti-mouse Alexa Fluor 568) or actin (rabbit anti–β-actin (1E35) directly conjugated with Alexa Fluor 488). Inset shows control cells treated with buffer/vehicle only. Scale bars on left and middle panels are 10 µm, and 5 µm on right panels. **(B)** PFA-fixed cells were stained with TUB2.1 and phalloidin–Alexa Fluor 488 to detect filamentous actin (F-actin). Boxed regions show cellular projections at increasing magnification and highlight overlapping patterns in actin and tubulin staining. Scale bars on left and middle panels are 10 µm, and 5 µm on right panels. **(C)** Kinesore-treated cells were briefly cold-treated before methanol fixation and antibody staining for actin and tubulin. Actin staining defines the projections in cold-treated cells. Scale bar is 10 µm.

intriguing possibility that a population of F-actin may reside within the microtubule lumen that is refractory to traditional means of detection, possibly due to limited antibody epitope/phalloidin binding site accessibility or conformation.

Supporting this proposition, a brief (60 s) cold treatment that disrupted tubulin staining enhanced actin antibody staining to the extent that actin staining defined the projection structure (Fig. 3 C).

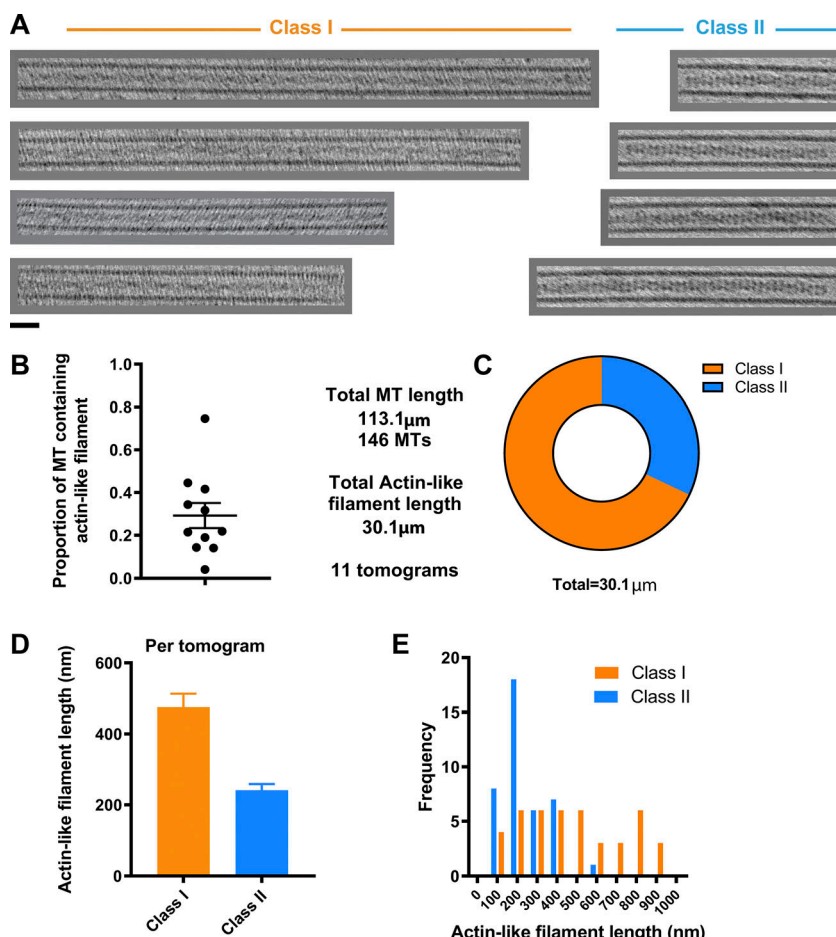

Figure 4. **Abundance and classification of two types of actin-like filament. (A)** Examples of lumenal filaments extracted from sub-volumes of 3D tomograms and projected in Z, chosen from straight sections of microtubule so that lumenal filaments can be shown over an extended length without being obscured by surrounding microtubule lattice. Images show projected sections of central regions of microtubules containing actin-like filaments and are separated into two groups (Class I and Class II) based on their morphology. Class II filaments were thicker and typically better defined. Scale bar is 25 nm. **(B)** Graph shows the proportion of microtubule lumen occupied by actin-like filaments (of either class) in 11 tomograms. **(C)** Proportion of actin-like filament lengths classified as Class I or Class II from the dataset. **(D and E)** Graph showing the average length of actin-like filaments (error bars represent SEM) of either class and their frequency distribution.

## Lumenal actin-like filaments have two distinct morphologies

A survey of the gross morphology of lumenal filaments resulted in their classification into two pools: Class I and Class II. Class I filaments are exemplified in the right orange box in Fig. 2 A, and Z-sections and class II by the left blue box. Further examples are shown in Fig. 4 A. Video 3 shows a Z-series through a microtubule containing an extended Class I filament adjacent to an "empty" microtubule. Class II filaments appeared slightly thicker and were typically better defined than Class I. Analysis of the tomograms in our dataset that provided clearest definition of lumenal contents (*n* = 11), which contained 146 microtubules with a total length of 113 μm, revealed that 27% of the total lumenal length of microtubule was occupied by actin-like filaments of either class (Fig. 4 B). There was considerable variation between tomograms with 4–76% of the lumenal length occupied, with a mean of 29%. Of the total actin-like filament length, 68% were Class I, and 32% were Class II filaments (Fig. 4 C). Although the frequencies of Class I and Class II filaments were similar, Class I filaments were longer (average, 475 nm) than Class II (average, 274 nm; Fig. 4, D and E). Microtubules containing Class I filaments had average outer and lumenal diameters of 25.56 ± 0.59 nm and 16.84 ± 0.76 nm, respectively. Those containing Class II filaments were typically slightly wider at 27.29 ± 0.58 nm (outer diameter) and 17.80 ± 0.85 nm (lumenal diameter), indicating that the presence of lumenal filaments either correlates with or modifies microtubule properties (Table S1). The

measured Class I diameters are close to those of a 13 protofilament microtubule (Zhang et al., 2018), which may suggest that Class II microtubules have an expansion in the lattice, or perhaps, more protofilaments.

## Layer-line analysis confirms that lumenal filaments are composed of F-actin

To further characterize microtubule cores, lumenal regions were extracted from the tomographic reconstructions as 3D sub-volumes. These were then summed in Z to obtain 2D projection images. Fourier transforms of these images were then calculated and inspected. Helical filaments like actin and tubulin display distinct layer-line patterns. Data from single representative microtubules containing filaments of the Class I and Class II varieties are shown in Fig. 5, and measured parameters from several microtubules/filaments are shown in Table S1. The layer lines we observe are highly consistent with prior analysis of the helical parameters of actin filaments extracted from tomograms of mammalian cells (Narita et al., 2012). Class I filaments display an actin layer-line pattern: a clear reflection at 5.94 ± 0.03 nm, the pitch of the short left-handed actin genetic helix (Actin Turn [L]), and a reflection at 29.53 ± 0.81 nm corresponding to the crossover spacing of the two right-handed long pitch actin helices (Actin Cross). This crossover spacing is shorter than for canonical actin (∼35 nm) but greater than that reported for actin cofilin filaments (∼27 nm; Hanson, 1967; McGough et al., 1997),

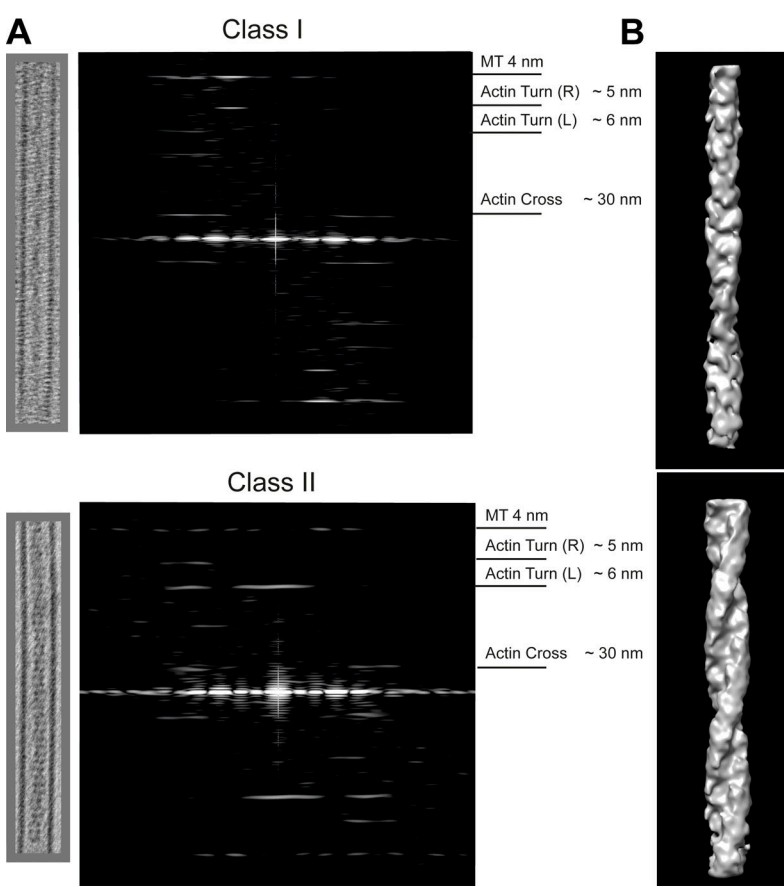

Figure 5. **Layer-line patterns of Class I and Class II filaments and 3D maps reveal typical actin repeating structures. (A)** Typical Class I (top) and Class II (bottom) lumenal filaments are extracted as 3D sub-volumes from the tomographic reconstruction and shown as projected in Z. The corresponding layer-line patterns are shown beside the projection. The characteristic 4-nm tubulin layer line was visible in all patterns. Parameters measured for more filaments are provided in Table S1. Class I filaments exhibited a clear reflection at 5.94 ± 0.03 nm, which is the pitch of the left-handed genetic helix of actin (Actin Turn [L]) and a reflection at 29.53 ± 0.81 nm corresponding to the crossover spacing of the right-handed long pitch helices (Actin Cross) as measured in the individual images. Reflections were observed in the Class II filaments at similar positions to Class I 6.11 ± 0.09 nm and 27.44 ± 2.28 nm, with an additional strong meridional reflection at 6.17 ± 0.06 nm (errors are SD). SEM is provided in Table S1 with measurements of additional actin reflection (Actin Turn [R]) at the pitch of the right-hand genetic helix. **(B)** 3D maps from representative extracted filament volumes using real space helical reconstructions with helical parameters derived from the models presented in Fig. S3 and Table S1. Maps are low pass filtered to 30 Å.

and actin is known to have a variable twist (Egelman et al., 1982). These results enable us to confidently identify the Class I lumenal filament as F-actin. Increased twist when compared with canonical actin may partly explain why phalloidin staining of this actin pool appears restricted and likely has implications for its detection with other probes (Kumari et al., 2019; McGough et al., 1997). A reflection at 4 nm was also visible as expected from the tubulin monomer axial repeat. Class II filaments also gave an actin layer-line pattern with reflections at 6.11 ± 0.09 nm (Actin Turn [L]) and 27.44 ± 2.28 nm (Actin Cross) that is augmented by a strong reflection on the meridian at 6.18 ± 0.06 nm (matching the pitch of the left-handed genetic helix of actin). This meridional reflection is indicative of additional proteins associated with the lumenal actin.

To support these conclusions, we note that the turn layer-line (L) for the Class I structure at 5.94 ± 0.03 nm compares well with the predicted spacing for the fifth layer line from a 29.5 nm actin helix repeat of 5.90 nm, whereas for the Class II structure, the observed spacing of 6.11 ± 0.09 nm compares well with 6.11 nm as predicted for the ninth layer line from a 2 × 27.5 nm actin axial repeat (Fig. S3, A and B). The radial positions of the observed layer-line peaks are also consistent with the expected radius of F-actin. Furthermore, both Class I and Class II structures gave reflections at the pitch of the right-handed actin genetic helix at 5.01 ± 0.04 nm and 4.99 ± 0.08 nm, respectively (Actin Turn [R]).

Using measured and modeled parameters, we generated real space helical reconstructions of Class I and Class II actin

filaments (Fig. 5 B). Class II filaments are wider at 10.4 nm vs. 8.9 nm for Class I, consistent with additional protein associated with the actin filament. We tentatively speculate that the meridional reflection we observed and this additional mass are consistent with a formin-like binding to the actin backbone (Gurel et al., 2014).

In summary, we have shown that microtubules can contain actin filaments within their lumen. We call these structures microtubule lumenal actin. We have distinguished two types of structure, Class I and Class II, with slightly different actin helical symmetries and outer microtubule diameters. The discovery of microtubule lumenal actin in our cell-based extrusion system prompts the questions of how it is incorporated, its functions, and its abundance in settings that are not small molecule–modified. Two hypotheses present themselves that are not mutually exclusive. First, altered microtubule dynamics caused by activation of kinesin-driven microtubule–microtubule sliding and bundling may provide an opportunity for actin to enter the microtubule lumen. This may occur through transient opening of the microtubule under strain when tight loops are formed (Fig. S1) or by kinesin-mediated disruption of the lattice (Coombes et al., 2016; Schaedel et al., 2019; Triclin et al., 2018). Put another way, F-actin incorporation may occur in response to mechanical or structural stress (Brangwynne et al., 2006), so a potential role for microtubule severing enzymes should also be considered (Vemu et al., 2018). Alternatively, microtubule extrusion has facilitated the analysis of a pool of preexisting actin-containing microtubules with previously limited accessibility by

high resolution cryo-EM. Indeed, the similarity of our images to the "dense core microtubules" first observed in amphibian and rat neurons as well as platelets is striking (Behnke and Zelander, 1967; Burton, 1984; Peters and Vaughn, 1967; Rodríguez Echandía et al., 1968; Xu and Afzelius, 1988). In either case, it will be important to understand if and how an F-actin core (of either class) alters the mechanical and dynamic properties of the microtubule, as well as explore this new concept as a new basis for actin-microtubule cross-talk in diverse settings (Dogterom and Koenderink, 2019).

## Materials and methods

### Cell culture and small molecule treatment

HAP1 cells were obtained from Horizon Discovery and cultured in Iscove's Modified Dulbecco's Medium containing 10% FCS and penicillin/streptomycin at 37°C in a 5% $CO_2$ incubator. For fluorescence imaging, cells were plated onto fibronectin-coated coverslips in a six-well plate at a density of $10^5$ per well the day before small molecule treatment. Cells were prepared for EM as described below. Kinesore (3,5-dibromo-N′-{[2,5-dimethyl-1-(3-nitrophenyl)-1H-pyrrol-3-yl]methylene}-4-hydroxy benzohydrazide) was obtained from Chembridge Corporation (6233307) and prepared as a 50 mM stock in DMSO. In addition to manufacturer-provided quality control, molecular weight was verified by mass spectrometry. To stimulate projection formation, treatments were performed in Ringer's buffer (155 mM NaCl, 5 mM KCl, 2 mM $CaCl_2$, 1 mM $MgCl_2$, 2 mM $NaH_2PO_4$, 10 mM glucose, and 10 mM Hepes, pH 6.8, in a 37°C incubator without $CO_2$) containing 100 µM kinesore (final DMSO concentration 0.2%) for 1 h. Control experiments were performed with DMSO alone in Ringer's buffer at 0.2%. Where indicated, cells were fixed with either ice-cold methanol or 4% PFA and stained for β-tubulin using the TUB 2.1 monoclonal antibody (1:1,000 dilution, T4026, Sigma-Aldrich). This was detected using a goat anti-mouse secondary antibody directly conjugated to Alexa Fluor 568 (1:400 dilution, A11004, Thermo Fisher Scientific). β-Actin was detected using the 13E5 rabbit monoclonal antibody directly conjugated to Alexa Fluor 488 (1:400 dilution, 8844, Cell Signaling Technology) or Alexa Fluor 488 phalloidin (1:400 dilution, A12379, Thermo Fisher Scientific). Where cold treatments were performed, dishes containing coverslips were placed in a shallow bath of ice-cold water for 60 s immediately before methanol fixation and processing as described.

### Light microscopy imaging and analysis

To label tubulin, HAP1 cells were incubated with 500 nM SiR-tubulin (Spirochrome) in growth media for 1 h before kinesore treatment as described above, before transfer to the microscope stage at 37°C. Images were acquired at 15 s intervals as single confocal sections, using a 633 nm laser on a Leica SP5-II system with a 63× objective lens. Fixed cells were imaged on the same system using 488 nm and 561 nm laser lines, were acquired as series of confocal sections, and are presented as maximum intensity projection images. Resulting data were further analyzed as indicated and prepared for publication using Fiji (ImageJ) and Adobe Photoshop/Illustrator Packages.

### CLEM

Cells were grown on Quantifoil R1.2/1.3 400 mesh gold EM grids (supplied by EM Resolutions Ltd.). 10-nm gold fiducial markers in BSA were added to the grids, and these were plunge-frozen in liquid ethane using a Leica EM GP plunge freezer and transferred to a Leica CryoCLEM stage based on the Leica DM6000FS fluorescence light microscope. In this microscope, the objective is cooled and the sample is transferred to a cryo stage where its temperature can be maintained below –140°C during observation, ensuring that the sample remains vitrified. Images were collected in both bright field and in green fluorescent channel. Areas of potential interest for further study by cryo transmission EM (cryoTEM) were thus recorded in a manner similar to Kukulski et al. (2011). The samples were recovered under liquid nitrogen and transferred to a Tecnai20 LaB6 TEM (FEI), operating at 200 kV using a Gatan 626 cryotransfer holder. The areas of interest were retraced and images collected on a bottom-mounted Thermo Fisher Scientific Ceta camera.

### Cryo-ET

Cryo samples were prepared as described above, clipped, and transferred into a Talos Arctica CryoTEM (FEI) operating at 200 kV. Tomographic series were acquired using a dose symmetric scheme between ±60° with images collected +20° to –60°, then +20° to +60° with angular increments of 3°, total dose of 109 e⁻/Å², and a defocus range between –2 µm and –4 µm. The acquisition magnification was 63,000 times, resulting in calibrated pixel size of 2.21 Å. Images were recorded in counted mode using a K2 Summit direct electron detector (Gatan) fitted to a BioQuantum energy loss spectrometer (Gatan) operating with a 20-eV slit width.

### Data processing

Reconstructions were performed using IMOD and its Etomo interface (University of Colorado, Boulder; Kremer et al., 1996). Briefly, image stacks containing bidirectional tilt-series data were imported into the software and subjected to coarse alignment using Tiltxcorr before fine fiducial-based alignment, positioning, and 2 × 2 binning. Tomograms were generated using the Weighted Back Projection with the Simultaneous Iterative Reconstruction Technique like filter to five iterations. The nominal voxel size of the final tomograms was 4.414 Å. Further processing and analysis were performed using the IMOD 3dmod interface to generate videos, extract sub-volumes, and perform microtubule/actin filament length analysis. Microtubules were modeled using 24-nm-diameter cylinder segments. The outer and lumenal diameters of the microtubules were measured on 2D projection images of extracted sub-volumes using Fiji. Images and sub-volumes were exported to Fiji or Adobe Photoshop to prepare for publication or further analysis as described below.

### Layer-line analysis

Fourier transforms of 2D projections of extracted filament volumes were calculated and the layer-line positions measured using Fiji (ImageJ). The well-characterized 4 nm reflection of the tubulin monomer was used as an internal calibration tool and the real space pixel size for each filament calculated. Modeling of

the layer-line patterns from helices with actin-like symmetry (Fig. S3) was performed using the HELIX program (Knupp and Squire, 2004; https://www.diamond.ac.uk/Instruments/Soft-Condensed-Matter/small-angle/SAXS-Software/CCP13/HELIX.html). A HELIX plug-in for ImageJ (https://imagej.nih.ij) can be obtained from c.knupp@gmail.com.

### Helical reconstruction

To directly compare the 3D structures of the two filament classes, real space helical reconstructions of individual filaments were calculated using EMAN2 (Tang et al., 2007). 85 nm long overlapping segments were extracted from 2D projections of representative filaments. The axial rise and subunit rotation determined from the modeling experiments and weighted back projection were used for the reconstruction. These representative 3D volumes were low-pass-filtered to 30 Å to minimize the high-frequency noise present in single-filament images.

### Online supplemental material

Fig. S1 shows live imaging of projection formation. Fig. S2 shows transverse sections through tomograms of several projections. Fig. S3 shows modeling and diffraction of actin filaments. Table S1 shows layer-line analysis and measurements of microtubule dimensions. Video 1 shows HAP1 cells labeled with SiR-tubulin and treated with kinesore. Video 2 shows a series of images through the tomogram described in detail in Fig. 2. Video 3 shows Z-series through a sub-volume containing two microtubules, one with a lumenal actin filament.

## Acknowledgments

We are grateful to Professor Carolyn Moores (Birkbeck, University of London) for helpful discussions on the project and comments on the manuscript draft.

This work was supported by the Biotechnology and Biosciences Research Council (BB/S000917/1) and a Lister Research Prize Fellowship to M.P. Dodding. D.M. Paul is supported by a British Heart Foundation Career Re-Entry Fellowship (FS/14/18/3071), which also supports J.M. Squire. D.M. Paul also has support from the Alan Turing Institute through a Turing Fellowship and the Academy of Medical Sciences by a Springboard award (SBF003/1142). J. Coombs was supported by the Engineering and Physical Sciences Research Council through the Bristol Centre for Functional Nanomaterials PhD programme. K.J. Surridge was supported by the Wellcome Trust four-year PhD in Dynamic Molecular Cell Biology programme. We acknowledge access and support of the GW4 Facility for High-Resolution Electron Cryo-Microscopy, funded by the Wellcome Trust (202904/Z/16/Z and 206181/Z/17/Z) and the Biotechnology and Biosciences Research Council (BB/R000484/1), as well as the Wolfson Bioimaging Facility at Bristol. The cryo-fluorescence microscope was supported by the Biotechnology and Biosciences Research Council (BB/L014181/1).

The authors declare no competing financial interests.

Author contributions: Investigation—D.M. Paul, J. Mantell, U. Borucu, J. Coombs, K.J. Surridge, P. Verkade, J.M. Squire, M.P. Dodding; Formal Analysis—D.M. Paul, J. Mantell, U. Borucu, J. Coombs, J.M. Squire, P. Verkade, M.P. Dodding; Writing—Original Draft—M.P. Dodding; Writing—Review and Editing—D.M. Paul, J.M. Squire, P. Verkade, M.P. Dodding; Funding Acquisition—D.M. Paul, P. Verkade, M.P. Dodding.

Submitted: 29 November 2019

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

# Supplemental material

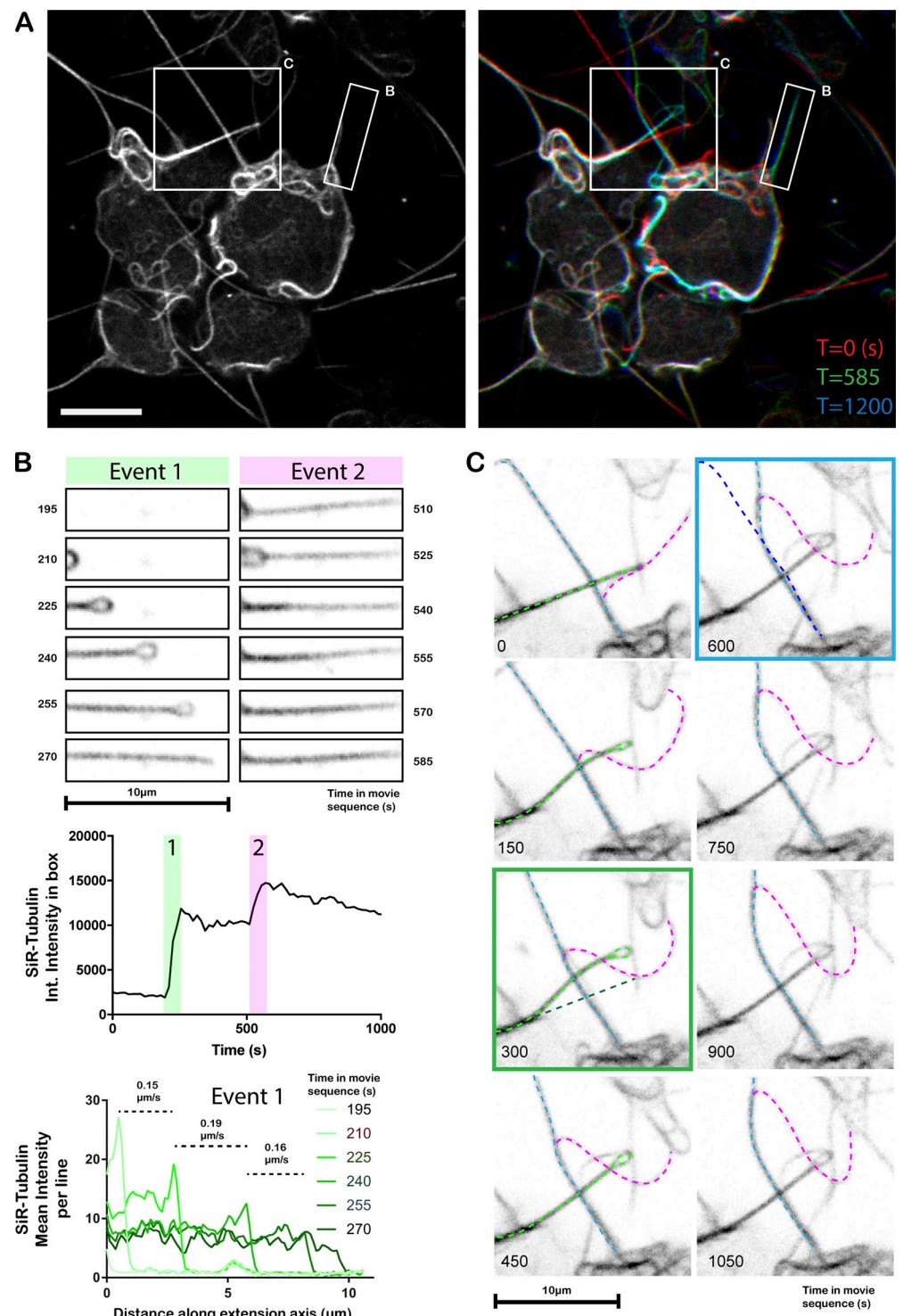

Figure S1. **Live-cell confocal imaging of microtubule projection formation through loop-extrusion in SiR-tubulin–labeled HAP1 cells. (A)** Left image shows cells at T = 0 s. Right image shows T = 0, T = 585, and T = 1,200 (time in s) in red, green, and blue channels, respectively. Boxes highlight areas that are expanded and analyzed in B and C. Scale bar, 10 µm. **(B)** Stills from the image series describing the formation and reinforcement of a microtubule-based projection through the extension/resolution of a loop. Green panels show extension of the first loop (event 1). Top graphs show change in tubulin integrated (Int.) intensity over the whole region as a function of time. The lower graph shows the average tubulin intensity per line as a function of distance along the axis of projection extension (x axis) and rate of extension. Magenta panels (event 2) show addition of further tubulin intensity over a similar time period later in the video, indicating that the projections are formed by the progressive layering of microtubules from the extrusion of loops. **(C)** Complex interactions between projections that affect their organization. Three extensions are highlighted by light blue, light green, and magenta dotted lines. The magenta projection moves along the light blue projection. The position of the light green filament is shifted as it passes. Green box panel shows original (dark green) and new positions (light green) of the projection. As the magenta proceeds along the blue filament, the blue projection is bent. Blue boxed panel shows original (dark blue) and new positions (light blue) of the blue projection.

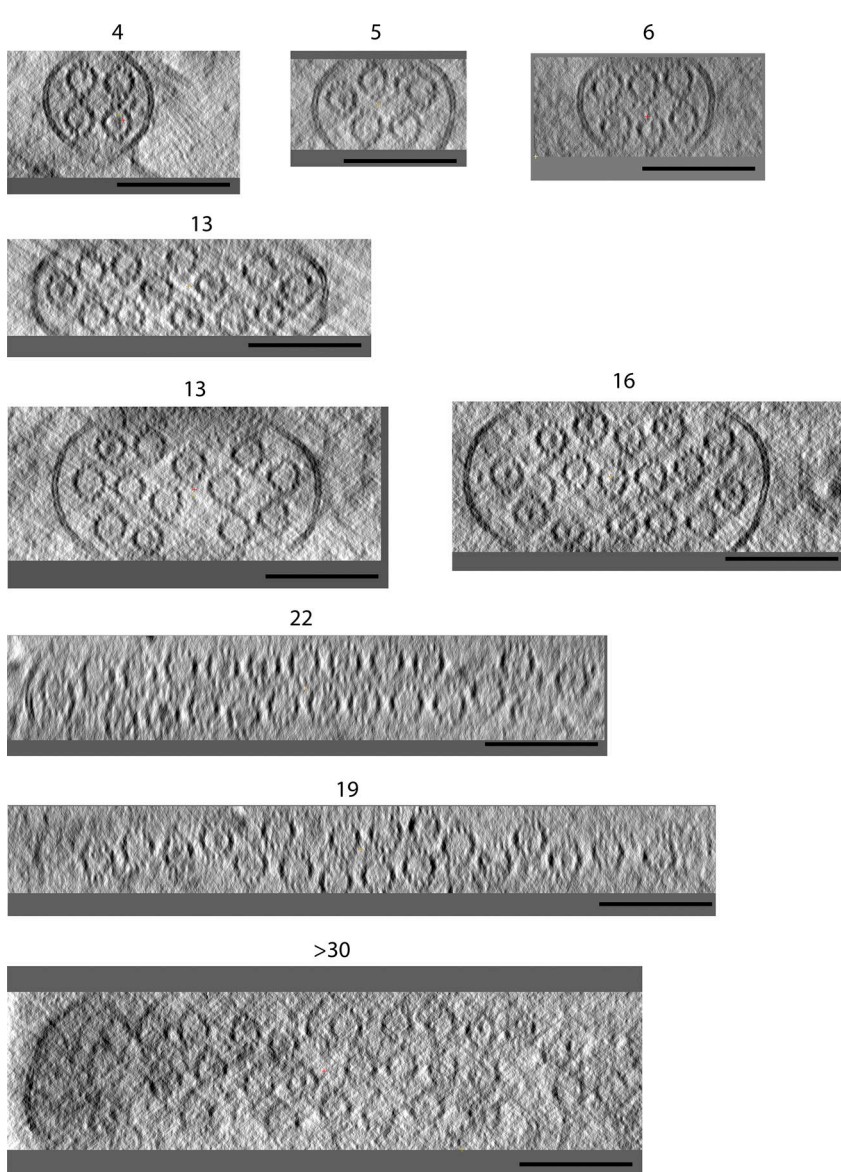

Figure S2.   **Sections through microtubule-based projections reveal variable microtubule number and close-packed organization.** Transverse sections through several microtubule-based projections show the close juxtaposition of variable numbers of microtubules. Lumenal density in many of the microtubules is apparent. Images are presented as summed slices. Scale bars, 100 nm.

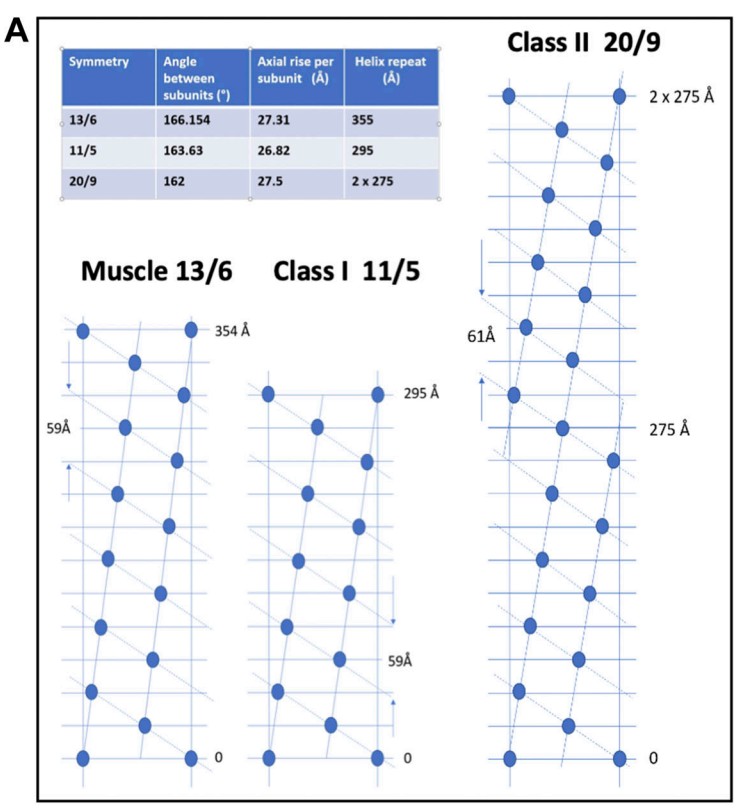

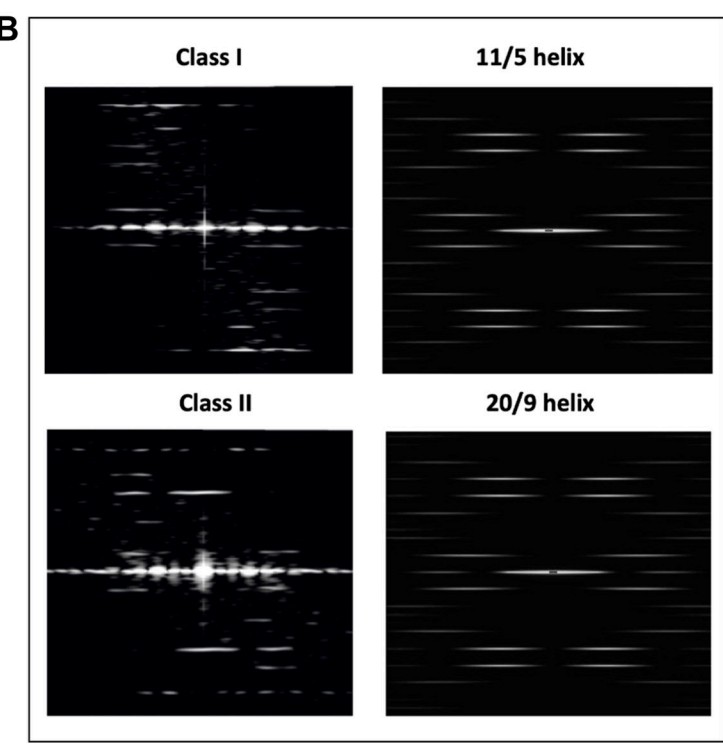

Figure S3. **Modeling and diffraction of actin filaments. (A)** Comparison of the radial nets of muscle actin filaments (13/6 helices of actin monomers) and Class I and Class II actin filaments as observed here, where Class I filaments have approximate 11/5 helical symmetry and Class II filaments have 20/9 helical symmetry. The inset table shows the helical parameters for the three symmetries. **(B)** Diffraction from Class I and Class II filaments and comparison to models. The images on the left reproduce the patterns shown in the main text Fig. 5 for Class I (top) and Class II (bottom) filaments. On the right are patterns produced by the HELIX program (Knupp and Squire, 2004) for actin-like filaments with 11/5 (top) and 20/9 (bottom) helical symmetry. A filament length of 100 subunits (simple spheres) was used with the monomer at 25 Å from the filament axis.

Video 1.   **Live-imaging of tubulin in kinesore-treated HAP1 cells.** Video shows HAP1 cells labeled with SiR-tubulin and treated with kinesore that is also shown in Fig. S1. Images were acquired every 15 s. Boxed region shows area of video described in more detail in Fig. 1.

Video 2.   **Tomographic analysis of microtubules within a projection.** Video shows a series of images through the tomogram described in detail in Fig. 1 B. Microtubules in the tomogram are then identified by colored cylinders.

Video 3.   **Z-series through a Class I filament–containing microtubule.** Video shows Z-series through a subvolume containing two microtubules. The top microtubule contains a Class I filament. The bottom microtubule contains globular densities.

**Table S1 is provided online as a separate Excel file. Table S1 shows measured parameters for individual Class I and Class II filaments. Microtubule dimensions and layer-line positions are listed for filaments of both classes. All data are shown in Angstroms. A comparison of observed (OBS) vs. calculated (CALC, from models) layer-line positions is also presented. N/A, not applicable as meridional layer line is not present in class I filaments.**

