## [Peer Review File · The Journal of Cell Biology]

In situ cryo-electron tomography reveals filamentous actin within the microtubule lumen

Danielle Paul, Judith Mantell, Ufuk Borucu, Jennifer Coombs, Katherine Surridge, John Squire, Paul Verkade, and Mark Dodding

Corresponding Author(s): Mark Dodding, University of Bristol and Paul Verkade, University of Bristol

Review Timeline:

Submission Date:	2019-11-29
Editorial Decision:	2019-12-26
Revision Received:	2020-03-16
Editorial Decision:	2020-04-23
Revision Received:	2020-05-06

Monitoring Editor: Eva Nogales

Scientific Editor: Marie Anne O'Donnell

Transaction Report:

DOI: <https://doi.org/10.1083/jcb.201911154>

December 26, 2019

Re: JCB manuscript #201911154

Dr. Mark P Dodding
University of Bristol
University Walk
London BS8 1TD
United Kingdom

Dear Dr. Dodding,

Thank you for submitting your manuscript entitled "In situ cryo-electron tomography reveals filamentous actin within the microtubule lumen". The manuscript was assessed by expert reviewers, whose comments are appended to this letter. We invite you to submit a revision if you can address the reviewers' key concerns, as outlined here.

As you will see from the reviewers, they think that your study is of interest and potential novelty but that the case should be strengthened concerning the identity of the filament in the microtubule lumen. Several suggestions are included. Although not all the proposals need to be included in the revised version of the manuscript, enough of them should be there to strengthen the actin claim without a doubt. We hope the detailed reviews are helpful and that you will consider resubmission to JCB.

GENERAL GUIDELINES:

Text limits: Character count for a Report is < 20,000, not including spaces. Count includes title page, abstract, introduction, results, discussion, acknowledgments, and figure legends. Count does not include materials and methods, references, tables, or supplemental legends.

Figures: Reports may have up to 5 main text figures. To avoid delays in production, figures must be prepared according to the policies outlined in our Instructions to Authors, under Data Presentation, <http://jcb.rupress.org/site/misc/ifora.xhtml>. All figures in accepted manuscripts will be screened prior to publication.

Supplemental information: There are strict limits on the allowable amount of supplemental data. Reports may have up to 3 supplemental figures. Up to 10 supplemental videos or flash animations are allowed. A summary of all supplemental material should appear at the end of the Materials and methods section.

Our typical timeframe for revisions is three months; if submitted within this timeframe, novelty will not be reassessed at the final decision. Please note that papers are generally considered through only one revision cycle, so any revised manuscript will likely be either accepted or rejected.

Thank you for this interesting contribution to Journal of Cell Biology. You can contact us at the journal office with any questions, cellbio@rockefeller.edu or call (212) 327-8588.

Sincerely,

Eva Nogales, Ph.D.
Monitoring Editor

Marie Anne O'Donnell, Ph.D.
Scientific Editor

Journal of Cell Biology

Reviewer #1 (Comments to the Authors (Required)):

In their study "In situ cryo-electron tomography reveals filamentous (F) actin within the microtubule lumen", Paul and colleagues study microtubule-rich protusions of HAP1 pharmacologically induced by treatment with the kinesin 1 modulator kinesore using cryo-ET. They find that a subpopulation of microtubules encase a second proteinaceous helical filament within their lumen, and provide evidence based on helical symmetry analysis that these filaments consist of 2 different conformers of F-actin, corroborated by fluorescence microscopy analysis. Finally, the authors speculate that this could be a more general phenomenon facilitating microtubule-actin crosstalk.

The authors demonstrate this phenomenon only in a single, highly artificial context. It is thus not yet at all clear that microtubules encasing another filament ever happens in a normal physiological context. However, I believe there is value in demonstrating this CAN happen, which is likely to stimulate further work in cell types and experimental contexts where it may be more difficult to observe. I also believe that it could encourage other researchers to "notice" this in their tomograms and report it, even if they were previously skeptical. I do not believe this study meets the bar of a normal JCB Article in terms of mechanistic insight. However, given the stated requirement of a more limited scope of the Report format in reporting striking results to open new avenues of research, I believe this study could potentially be appropriate as a report should its conclusions prove justified.

As the main conclusion of the paper is that F-actin can be found in the microtubule lumen, I believe this single point needs to be proven beyond a reasonable doubt. While the authors have provided evidence in support of this, which seems the most parsimonious explanation, I believe this should be strengthened in order for this study to be in principle suitable for acceptance. I thus do not support

acceptance unless the following major points are addressed:

Major issue: proving the luminal filament is F-actin

1) Layer line analysis

It is quite clear from the authors' data that the contents of the microtubules are 2 stranded protein filaments which are morphologically consistent with F-actin. However, the helical layer line analysis performed by the authors is somewhat confusing. To analyze the helical symmetry of the filaments, the authors project their tomograms, then perform helical layer-line analysis on the power spectra of the Fourier transforms.

-They note the appearance of a layer line at 59.4 Angstroms for class 1, and 61.1 Angstroms for class 2, which they say is characteristic of the "genetic" helix of F-actin. While this does indeed almost correspond to the standard 54 Angstrom axial spacing of the actin protomer in the "long pitch", shallow right-handed actin helix, it is unclear why there is not a layer line at ~27 Angstroms for both classes, the axial spacing of the short pitch left-handed helix which is normally observed. Is the resolution of the data intrinsically too poor to observe this layer line? The methods section is insufficiently detailed to determine if this need be the case, as the authors do not state at what magnification the data were collected, or the voxel size of the final reconstructions. This information should be included.

-If the authors cannot feasibly observe this layer-line in the projected tomograms, one approach would be to take a few high-dose, high-magnification projection images of microtubule-rich protrusions (of the type typically used for single-particle analysis) for layer-line analysis. Observation of the 27 Angstrom layer line would make their argument substantially more compelling.

-What is the source of the strong meridional reflection in Class II which does not at all match the simulated pattern (Supplementary Figure 6)? The authors speculate this comes from an encircling formin. Are the Class II filaments consistently thicker than Class I, which is a necessary correlate of this speculation? Otherwise, this is difficult to reconcile.

2) Corroborating evidence for the presence of F-actin

The fluorescence microscopy analysis provided (Supplementary Figure 4) is moderately convincing, but there is not extensive overlap between the tubulin and F-actin signals. One reasonable possibility, as the authors note, is that the luminal F-actin is inaccessible to F-actin labelling reagents such as phalloidin. The study would overall be much more convincing if the authors could provide stronger independent molecular evidence (not simply based on symmetry analysis of the EM data) that the luminal filaments are indeed composed of F-actin.

-One suggestion would be to pharmacologically depolymerize the microtubules and see if more F-actin could be labelled with phalloidin, which would be consistent with the phalloidin inaccessibility model. I.e. sequentially treat cells with HAP1, then colchicine, then fix and stain with phalloidin.

-A second suggestion would be to pre-treat the cells with a drug which selectively prevents actin polymerization (e.g. latrunculin A), followed by kinesore, then perform cryo-ET on protrusions. If this caused disappearance of the encased filaments in cryo-ET, their identity as actin filaments would be very well-supported.

-Other approaches, e.g. mechanically isolating the protusions by shake-off for protein composition analysis by mass spectrometry or quantitative western blotting to determine the stoichiometries of tubulin and actin, could also be convincing and would be welcome additions. However, I realize this is likely to be beyond the scope of work feasible for a revision.

Minor issues:

#1) Typo on line #34 "for cyro", which should be fixed to "for cryo".

#2) Typo on line #40 "formed though", which should be fixed to "formed through".

Reviewer #2 (Comments to the Authors (Required)):

This short report presents an interesting discovery that extended segments of F-actin filaments can present in microtubule lumen of kinesore-induced microtubule bundles of cellular projections. The data quality of cryo-electron tomographic analysis is excellent and enabled the authors to characterize the helical filaments in the lumen of the microtubules. The densities in the lumen display the structural features of F-actin filaments. Further antibody labeling results agree with the cryo-ET finding. Proteins present in the lumen of microtubules have been reported decades ago, but most of their identities are unclear. This is the first time that actin filaments are reported being found in the lumen of microtubules of dynamic cell projections. Microtubule and actin filaments are the two major dynamic cytoskeleton components in cells. Knowledge on co-plays between the two is fundamental in cell biology. Although the biological significance of this phenomenon is unclear at the moment, this novel case report is anticipated to attract further investigation and may lead to novel hypotheses for new findings. Therefore, I suggest the manuscript be accepted for publication as a short report after a revision following the comments below.

(1) The significant point for cell biology in this short report is that there are thin (class I) and thick (class II) F-actin filaments present in the lumen of microtubules of the dynamic cell projections kinesore induced. Fig. 3 demonstrates for this conclusion with structural analysis, and the supplementary Fig. S4 supports the conclusion with fluorescence labeling. The Fig. S4 should be included in the main text. The authors may consider to merge the Fig.2A and Fig.3 together as a new Fig 2A. If the space is a concern, Fig. 2B-2E can be moved to Supplementary materials and the population statistics of each types (F-actins and associated microtubules) can be summarized in a simple table. Then the supplementary Fig. S4 can be included in the main text as the new Fig.3.

(2) The report also analyzed the microtubules with Class I and Class II filaments inside and demonstrates differences in inter-and outer-diameters. How about the diameters of the microtubules without any filaments inside (call it the class 0)? It will be informative to also include the empty microtubules in the same cryo-tomograms in the comparison.

(3) Page 3, Line 34: "...cyro Correlative Light Electron Microscopy..." should be "...cryo Correlative Light Electron Microscopy...".

(4) Page 5, Line 125: "...reflection is indicative an additional protein(s)..." should be "...reflection is indicative of additional protein(s)..."

(5) Page 7, Line 183: "HAP1 cells were obtained Horizon Discovery..." should be "HAP1 cells were obtained from Horizon Discovery...".

(6) Page 7, Line 192: "...stimulate projection formation." should be "...stimulate projection formation, ", i.e., a comma instead of a period.

Reviewer #3 (Comments to the Authors (Required)):

This manuscript by Paul et al. presents a very interesting discovery of filamentous actin inside the microtubule lumen. Although this phenomenon was currently observed in small molecule induced microtubule based cellular projections, there is a good chance that such phenomenon also exists in normal cellular context. Given its novelty and potential impact on the entire cell biology field, I highly recommend publishing this manuscript in JCB after minor revisions (mostly about the organization of the figures).

1) I assume this manuscript was submitted as "Reports", which allows up to 5 figures. Therefore, I don't understand why the authors don't move some of the supplemental figures to the main figures. Good candidates are Fig S2, S3 and S4, especially Fig S4. The current Fig 1 is too compact and overwhelming, and can be split into two main figures. The current Fig 3 is too technical for the general readers and therefore should be moved to supplements.

2) Cryo-ET is the main structural technique used in this study, however, throughout the manuscript, I don't see any 3D structure showing actin filament inside microtubule. The authors tried to use the comparison of layer line profile to demonstrate the similarity between the luminal filaments and actin filaments. However, in my opinion, a much better approach is to do direct side-by-side comparison of their 3D structures low-pass filtered to a similar resolution.

3) Since no structural study was done in the previous paper (Randall 2017 PNAS), the cryo-ET data presented in this manuscript provides great insight of the molecular mechanism underlying the phenotype of cellular projections formation upon kinesore treatment, which to me is equally interesting to the discovery of luminal actin filament. The authors mentioned that "microtubules typically maintained a consistent spacing of between 10-25 nm". How does this distance compare to the size of kinesin-1 in its activated form, and other microtubule cross-linkers such as PRC1?

4) In the Discussion, the author stated that "we also note the presence of an actin subunit discovered within the lumen of the γ -tubulin ring complex (the major microtubule nucleator) in two recent studies. This suggests one could begin to consider a co-nucleation model." Although I understand one is allowed to speculate a bit in the discussion, I am not sure I agree with this hand-waving model. Actin or actin-like protein were also found as core components of dynactin complex, and at the base of inner dynein arms of axoneme. I think actin monomer or oligomer are there to play a structural role.

Response to Reviewers

We are very grateful for the positive and constructive comments from all of the reviewers that have helped to improve the presentation of our manuscript and provide important reinforcement of the key conclusions of the study. Our response to those comments is provided on the document below in green. Relevant modifications to the text in response to these comments are highlighted in purple.

Reviewer #1 (Comments to the Authors (Required)):

In their study "In situ cryo-electron tomography reveals filamentous (F) actin within the microtubule lumen", Paul and colleagues study microtubule-rich protrusions of HAP1 pharmacologically induced by treatment with the kinesin 1 modulator kinesore using cryo-ET. They find that a subpopulation of microtubules encase a second proteinaceous helical filament within their lumen, and provide evidence based on helical symmetry analysis that these filaments consist of 2 different conformers of F-actin, corroborated by fluorescence microscopy analysis. Finally, the authors speculate that this could be a more general phenomenon facilitating microtubule-actin crosstalk.

The authors demonstrate this phenomenon only in a single, highly artificial context. It is thus not yet at all clear that microtubules encasing another filament ever happens in a normal physiological context. However, I believe there is value in demonstrating this CAN happen, which is likely to stimulate further work in cell types and experimental contexts where it may be more difficult to observe. I also believe that it could encourage other researchers to "notice" this in their tomograms and report it, even if they were previously skeptical. I do not believe this study meets the bar of a normal JCB Article in terms of mechanistic insight. However, given the stated requirement of a more limited scope of the Report format in reporting striking results to open new avenues of research, I believe this study could potentially be appropriate as a report should its conclusions prove justified.

As the main conclusion of the paper is that F-actin can be found in the microtubule lumen, I believe this single point needs to be proven beyond a reasonable doubt. While the authors have provided evidence in support of this, which seems the most parsimonious explanation, I believe this should be strengthened in order for this study to be in principle suitable for acceptance. I thus do not support acceptance unless the following major points are addressed:

Major issue: proving the luminal filament is F-actin

1) Layer line analysis

It is quite clear from the authors' data that the contents of the microtubules are 2 stranded protein filaments which are morphologically consistent with F-actin. However, the helical layer line analysis performed by the authors is somewhat confusing. To analyze the helical symmetry of the filaments, the authors project their tomograms, then perform helical layer-line analysis on the power spectra of the Fourier transforms.

-They note the appearance of a layer line at 59.4 Angstroms for class 1, and 61.1 Angstroms for class 2, which they say is characteristic of the "genetic" helix of F-actin. While this does indeed almost correspond to the standard 54 Angstrom axial spacing of the actin protomer in the "long pitch", shallow right-handed actin helix, it is unclear why there is not a layer line at

~27 Angstroms for both classes, the axial spacing of the short pitch left-handed helix which is normally observed. Is the resolution of the data intrinsically too poor to observe this layer line? The methods section is insufficiently detailed to determine if this need be the case, as the authors do not state at what magnification the data were collected, or the voxel size of the final reconstructions. This information should be included.

The methods have been expanded to include the information requested. Please see below for main response.

-If the authors cannot feasibly observe this layer-line in the projected tomograms, one approach would be to take a few high-dose, high-magnification projection images of microtubule-rich protrusions (of the type typically used for single-particle analysis) for layer-line analysis. Observation of the 27 Angstrom layer line would make their argument substantially more compelling.

Taking the above two points together, and before moving on to describe new data, we would like to clarify our interpretation of the data as originally presented. We attribute the 59.4/61.1 Angstrom layer-lines (for Class I/Class II filaments respectively) to the pitch of the short left-handed 'genetic' helix, *not* the axial spacing of the protomer in the right-handed long-pitch helix. The typical diffraction pattern from actin filaments does not have a reflection at 54/55 Angstroms, even though this is the axial spacing along the long-pitch helices. The usual patterns have two near-meridional layer-line peaks, one at the pitch of the left-hand genetic helix at around 59 Angstroms and the other at the pitch of the right-hand genetic helix which is around 51 Angstroms. For reference, see figure on the *in situ* tomography of actin filaments by Narita *et al.* JMB 2012 on the right that is now also cited in the manuscript.

Nothing is usually observed between these two layer-lines. In addition, as the reviewer notes, actin structures would be expected to give a relatively weak meridional peak at around 27 Angstroms at the axial spacing of the short pitch helix.

Considered on this basis, the match to the classical actin filament is almost exact for the data as first presented: for Class I filaments of crossover repeat 295 Angstroms we see a near-meridional layer-line at 59.36 (+/- 0.31) Angstroms – predicted spacing $295/5 = 59.0$ Angstroms; for Class II filaments of crossover repeat 275 Angstroms it is seen at 61.12 (+/- 0.86) Angstroms – predicted spacing $2 \times 275/9 = 61.11$ Angstroms.

In addition, the radius of the diffracting object can be determined from the distance of these peaks from the meridian (central vertical axis). The observed X-ray diffraction patterns from actin filaments are fitted quite well with actin monomer centres at a radius of around 25 Angstroms from the filament axis. This is the value that was used to calculate the model

diffraction patterns shown in **Figure S3**. The observed peaks are at a similar radius to those predicted. This is now made more explicit in the text with the statement ‘The radial positions of the observed layer line peaks are also consistent with the expected radius of F-actin’

In summary, the parameters we originally described almost *perfectly* match those which would be predicted from actin filaments.

Nonetheless, we appreciate the reviewer’s point that visualisation of the 27 Angstrom layer line from the axial spacing of the short helix would be highly satisfactory and provide important confirmation if it could be observed. Further support would also be provided by identification of the 51 Angstrom layer-line from the pitch of the right-handed ‘genetic’ helix. These new measurements are now provided in Table S1. We observed a near 51 Angstrom layer line in all of the Class I and Class II filaments examined, that was also present in the Fourier transforms presented in our first manuscript although not annotated. For Class I filaments this is observed at 50.05 (+/- 0.39) Angstroms. For Class II filaments this is observed at 49.87 (+/- 0.75) Angstroms. We also observe a weak meridional layer line in all of the Class I filaments at 27.55 (+/- 0.59) Angstroms and 8 of 18 Class II filaments at 26.64 (+/- 0.92) Angstroms. Importantly, these values are in good agreement with those predicted from our helical models and OBS:CALC ratios are presented with this new data in Table S1 and Figure S3.

Together, these measurements now specify the crossover of the long pitch helices, the left-handed ‘genetic’ helical pitch, the right-handed ‘genetic’ helical pitch and the protomer axial spacing. All are a near perfect match for F-actin and we suggest that this now provides unambiguous *in situ* identification of the filaments.

-What is the source of the strong meridional reflection in Class II which does not at all match the simulated pattern (Supplementary Figure 6)? The authors speculate this comes from an encircling formin. Are the Class II filaments consistently thicker than Class I, which is a necessary correlate of this speculation? Otherwise, this is difficult to reconcile.

To address the reviewer’s comment, and also the comment of reviewer 3, we have now calculated 3D maps of Class I and Class II filaments using real space helical reconstruction with helical parameters from our models. These maps are presented filtered to 30 Angstrom resolution. These new data are shown in Figure 5B. It is clear from these images that Class II filaments, which contain the extra density, are thicker than Class I. We have measured this as 1.17x (8.9 nm for Class II and 10.4 nm for Class I).

2) Corroborating evidence for the presence of F-actin

The fluorescence microscopy analysis provided (Supplementary Figure 4) is moderately convincing, but there is not extensive overlap between the tubulin and F-actin signals. One reasonable possibility, as the authors note, is that the luminal F-actin is inaccessible to F-actin labelling reagents such as phalloidin. The study would overall be much more convincing if the authors could provide stronger independent molecular evidence (not simply based on symmetry analysis of the EM data) that the luminal filaments are indeed composed of F-actin.

-One suggestion would be to pharmacologically depolymerize the microtubules and see if more F-actin could be labelled with phalloidin, which would be consistent with the phalloidin inaccessibility model. I.e. sequentially treat cells with HAP1, then colchicine, then fix and stain with phalloidin.

-A second suggestion would be to pre-treat the cells with a drug which selectively prevents actin polymerization (e.g. latrunculin A), followed by kinesore, then perform cryo-ET on protusions. If this caused disappearance of the encased filaments in cryo-ET, their identity as actin filaments would be very well-supported.

-Other approaches, e.g. mechanically isolating the protusions by shake-off for protein composition analysis by mass spectrometry or quantitative western blotting to determine the stoichiometries of tubulin and actin, could also be convincing and would be welcome additions. However, I realize this is likely to be beyond the scope of work feasible for a revision.

We thank the reviewer for suggesting several strategies by which we can improve the corroborating evidence. We have focussed on the first suggestion although with some modification in the approach. To address the reviewer's point that if microtubule structure were disrupted, actin staining should be enhanced, we choose to use cold shock rather than an additional small molecule to acutely disrupt the projection microtubule structure. After some optimisation, we found that a brief (60 second) cold treatment disrupted tubulin staining with the projections. Concomitantly, this resulted in an increase in actin antibody staining, which now essentially defines the projection structure and appears consistent with the quite high abundance (around 30% occupancy) of filaments in the microtubule lumen observed in our EM analysis. These data are now presented in Figure 3C. Combined with the PFA fixation/phalloidin staining experiment presented in Figure 3B, this provides evidence for a pool of actin/F-actin that is refractory to detection when microtubules are intact, but is amenable for staining by both antibodies and phalloidin when microtubules are disrupted.

We are grateful for the reviewer's constructive comments that have prompted us to substantially improve the data supporting the main claim of the paper; we suggest that there can now be no reasonable doubt that the luminal filaments are composed of F-actin.

Minor issues:

#1) Typo on line #34 "for cyro", which should be fixed to "for cryo".

#2) Typo on line #40 "formed though", which should be fixed to "formed through".

We thank the reviewer for identifying these typos and we have corrected them in the revised manuscript.

Reviewer #2 (Comments to the Authors (Required)):

This short report presents an interesting discovery that extended segments of F-actin filaments can present in microtubule lumen of kinesore-induced microtubule bundles of cellular projections. The data quality of cryo-electron tomographic analysis is excellent and enabled the authors to characterize the helical filaments in the lumen of the microtubules. The densities in the lumen display the structural features of F-actin filaments. Further antibody labeling results agree with the cryo-ET finding. Proteins present in the lumen of microtubules have been reported decades ago, but most of their identities are unclear. This is the first time that actin filaments are reported being found in the lumen of microtubules of dynamic cell projections. Microtubule and actin filaments are the two major dynamic cytoskeleton components in cells. Knowledge on co-plays between the two is fundamental in cell biology. Although the

biological significance of this phenomenon is unclear at the moment, this novel case report is anticipated to attract further investigation and may lead to novel hypotheses for new findings. Therefore, I suggest the manuscript be accepted for publication as a short report after a revision following the comments below.

(1) The significant point for cell biology in this short report is that there are thin (class I) and thick (class II) F-actin filaments present in the lumen of microtubules of the dynamic cell projections kinesore induced. Fig. 3 demonstrates for this conclusion with structural analysis, and the supplementary Fig. S4 supports the conclusion with fluorescence labeling. The Fig. S4 should be included in the main text. The authors may consider to merge the Fig.2A and Fig.3 together as a new Fig 2A. If the space is a concern, Fig. 2B-2E can be moved to Supplementary materials and the population statistics of each types (F-actins and associated microtubules) can be summarized in a simple table. Then the supplementary Fig. S4 can be included in the main text as the new Fig.3.

We appreciate the reviewer's suggestions to improve the presentation of our manuscript. The structure of the manuscript (figures only) has now been extensively revised to take full advantage of the 5 figures offered by the JCB report format. Amongst other changes also suggested by reviewers 1 and 3, the previous Fig S4 is now presented as an expanded Figure 3. The number of supplementary figures has been reduced to 3 as required by the report format.

(2) The report also analyzed the microtubules with Class I and Class II filaments inside and demonstrates differences in inter-and outer-diameters. How about the diameters of the microtubules without any filaments inside (call it the class 0)? It will be informative to also include the empty microtubules in the same cryo-tomograms in the comparison.

A note directing the reader to a side-by-side comparison of a Class I and an empty microtubule in Video 3 has been added to the text, reading 'Video 3 shows a Z-series through a microtubule containing an extended Class I filament adjacent to an 'empty' microtubule.' We have added an additional line to the text discussing this in the context of typical microtubule diameters reading 'The different microtubule diameters of the Class I and Class II filament containing structures suggest the possibility that there may be different microtubule protofilament numbers associated with the two classes. The measured Class I diameters are close to those of a 13 protofilament microtubule (Zhang et al., 2018) which may suggest that Class II filament containing microtubules have 14 protofilaments or an expansion in the 13 protofilament lattice'. We would prefer not to directly report a formal 'empty' microtubule diameter at this stage because the frequency of luminal filaments is such that they may be adjacent to 'empty' sections of microtubule either with or outside of the tomogram and influence this analysis. We hope that the reviewer considers the comparison to literature for the measurements we can definitively make as sufficient.

(3) Page 3, Line 34: "...cryo Correlative Light Electron Microscopy..." should be "...cryo Correlative Light Electron Microscopy...".

(4) Page 5, Line 125: "...reflection is indicative an additional protein(s)..." should be "...reflection is indicative of additional protein(s)..."

(5) Page 7, Line 183: "HAP1 cells were obtained Horizon Discovery..." should be "HAP1 cells were obtained from Horizon Discovery..."

(6) Page 7, Line 192: "...stimulate projection formation." should be "...stimulate projection formation, ", i.e., a comma instead of a period.

We thank the reviewer for identifying these typos and we have corrected them in the revised manuscript.

Reviewer #3 (Comments to the Authors (Required)):

This manuscript by Paul et al. presents a very interesting discovery of filamentous actin inside the microtubule lumen. Although this phenomenon was currently observed in small molecule induced microtubule based cellular projections, there is a good chance that such phenomenon also exists in normal cellular context. Given its novelty and potential impact on the entire cell biology field, I highly recommend publishing this manuscript in JCB after minor revisions (mostly about the organization of the figures).

1) I assume this manuscript was submitted as "Reports", which allows up to 5 figures. Therefore, I don't understand why the authors don't move some of the supplemental figures to the main figures. Good candidates are Fig S2, S3 and S4, especially Fig S4. The current Fig 1 is too compact and overwhelming, and can be split into two main figures. The current Fig 3 is too technical for the general readers and therefore should be moved to supplements.

As also described in our response to reviewer 2, the figure presentation has been extensively revised to take full advantage of the 5 figures offered by JCB as a report. Briefly, the previous figure 1 has now been split to separate CLEM data from the tomography as suggested. They are now presented in the revised manuscript as Figure 1 and 2. The previous Fig. S2 has been incorporated into the revised Figure 1. The previous Fig. S4 has been moved into the main text (now Figure 3) and expanded in response to the comments of reviewer 1.

We agree that Fourier transform figure (previously Figure 3, now Figure 5) is quite technical in its analysis and interpretation. However, it is not uncommon to present such data in this manner and we think that the underlying concepts should be accessible to a general reader. In particular, the points that the pattern are superimpositions of both tubulin and actin periodic features, and that the Class I and Class II filaments differ mainly by the presence of the additional meridional reflection in Class II filaments are straightforward and essential pieces of data that support the main conclusions of the study. To the best of our knowledge, a striking pattern like this incorporating both tubulin and actin features in this way has never been shown before and our strong preference is to retain it as a main figure. However, we have modified the figure with clearer annotations to highlight the Actin Turn(L), Actin Turn (R) and crossover features and now provide 3D maps alongside which provides better context (see below).

2) Cryo-ET is the main structural technique used in this study, however, throughout the manuscript, I don't see any 3D structure showing actin filament inside microtubule. The authors tried to use the comparison of layer line profile to demonstrate the similarity between the luminal filaments and actin filaments. However, in my opinion, a much better approach is to do direct side-by-side comparison of their 3D structures low-pass filtered to a similar resolution.

To address the reviewer's comment, and also the comment of reviewer 1, we have now calculated 3D maps of Class I and Class II filaments using real space helical reconstruction using helical parameters from our modelling. These maps are presented as low-pass filtered to 30 Angstrom resolution. These new data are shown in Figure 5B.

3) Since no structural study was done in the previous paper (Randall 2017 PNAS), the cryo-

ET data presented in this manuscript provides great insight of the molecular mechanism underlying the phenotype of cellular projections formation upon kinesore treatment, which to me is equally interesting to the discovery of luminal actin filament. The authors mentioned that "microtubules typically maintained a consistent spacing of between 10-25 nm". How does this distance compare to the size of kinesin-1 in its activated form, and other microtubule cross-linkers such as PRC1?

The reviewer raises an important point. We also believe that this assay system has potential for the understanding of kinesin-1 activation and the size/structure of its active form. The MT-MT spacing observed here is of interest.

As noted in the original draft, the spacing observed here is consistent with the *in vitro* study from Andrews *et al.* (PNAS 1993) using purified kinesin and microtubules, where microtubules were observed to be linked by 'cross-bridges typically $< \text{or} = 25\text{nm}$ long'. It is also consistent with the more recent fluorescence interference contrast microscopy study from Kerssemakers *et al.* (PNAS 2006) that again *in vitro*, kinesin-1 'elevates gliding microtubules $17 \pm 2 \text{ nm}$ above a surface'.

Going forward, we hope that this system will help us to resolve the long-standing question of what conformation(s) kinesin-1 assumes in its active state that we think is likely to be substantially more compact than that which is typically represented in animations, cartoons and schematics.

For the present manuscript, we have reinforced the point on spacing and consistency with other studies by citing the additional Kerssemakers *et al.* reference with new text now reading,

'Within these structures, microtubules typically maintained a consistent spacing of between 10-25 nm (blue shading) although some were also observed to traverse bundles (yellow shading). This spacing is consistent with *in vitro* EM studies of kinesin mediated microtubule-microtubule cross-linking ($\leq 25 \text{ nm}$) (Andrews *et al.*, 1993) and measurements of the distance kinesin-1 holds its cargoes from the microtubule surface ($\approx 17\text{nm}$) (Kerssemakers *et al.*, 2006).'

4) In the Discussion, the author stated that "we also note the presence of an actin subunit discovered within the lumen of the γ -tubulin ring complex (the major microtubule nucleator) in two recent studies. This suggests one could begin to consider a co-nucleation model." Although I understand one is allowed to speculate a bit in the discussion, I am not sure I agree with this hand-waving model. Actin or actin-like protein were also found as core components of dynactin complex, and at the base of inner dynein arms of axoneme. I think actin monomer or oligomer are there to play a structural role.

We take the reviewer's point that this is over-speculative and have removed this from the revised manuscript.

April 23, 2020

RE: JCB Manuscript #201911154R

Dr. Mark P Dodding
University of Bristol
University Walk
London BS8 1TD
United Kingdom

Dear Dr. Dodding:

Thank you for submitting your revised manuscript entitled "In situ cryo-electron tomography reveals filamentous actin within the microtubule lumen". We would be happy to publish your paper in JCB provided the text is amended to address Rev#1 and #2's remaining points where you consider appropriate, and pending final revisions necessary to meet our formatting guidelines (see details below).

- Provide the main and supplementary texts as separate, editable .doc or .docx files
- Provide main and supplementary figures as separate, editable files according to the instructions for authors on JCB's website paying particular attention to the guidelines for preparing images and blots at sufficient resolution for screening and production
- Format references for JCB
- Videos and tables should be include in the Online Supplementary Material paragraph
- Provide tables as excel files
- Add scale bars to figures 2A (if not all same scale), 3A, B (panels on right / zoom), 4A?, 5A?

A. MANUSCRIPT ORGANIZATION AND FORMATTING:

Full guidelines are available on our Instructions for Authors page, <http://jcb.rupress.org/submission-guidelines#revised>. **Submission of a paper that does not conform to JCB guidelines will delay the acceptance of your manuscript.**

B. FINAL FILES:

- An editable version of the final text (.DOC or .DOCX) is needed for copyediting (no PDFs).
- High-resolution figure and video files: See our detailed guidelines for preparing your production-

ready images, <http://jcb.rupress.org/fig-vid-guidelines>.

Thank you for this interesting contribution, we look forward to publishing your paper in Journal of Cell Biology.

Sincerely,

Eva Nogales, Ph.D.
Monitoring Editor

Marie Anne O'Donnell, Ph.D.
Scientific Editor

Journal of Cell Biology

Reviewer #1 (Comments to the Authors (Required)):

In their revised manuscript, Paul et al. have addressed my concerns and present a substantially strengthened story. The cold-shock experiment to depolymerize microtubules and reveal further actin staining in the kinesore-induced cellular projections is convincing additional evidence. Furthermore, the presented reconstructions of filaments are very welcome and intuitive, and the expanded methods section regarding the tomography is very thorough. I have a few very minor comments detailed below, and I believe the MS should be accepted with textual revisions responding to these comments without further delay.

Minor concerns:

-27 Angstrom layer lines:

Presumably in response to my previous comments, the authors have now included detailed views of claimed 27 Angstrom layer lines in Figure S3 (yellow arrows). These really are not very convincing, and seem to have similar intensity to the background noise. As including them frankly weakens the authors' case, I recommend removing this from Figure S3 and from the text. It is perfectly reasonable that the tomograms be too low resolution to resolve this feature, and this can be simply stated if the authors so desire.

-Narita JMB 2012 reference:

Including the reference to the previous work by Narita and colleagues is very helpful and convincing, particularly because this analysis was performed in a similar experimental context (filaments extracted from tomograms of mammalian cells). While very well-explained in the rebuttal letter, this paper is currently referenced in a funny way in the actual manuscript, grouped together with McGough et al. regarding the helical parameters of cofilactin.

I believe this merits its own sentence for clarity, something along the lines of "Furthermore, the layer lines we observe are highly consistent with prior analysis of the helical parameters of actin filaments extracted from tomograms of mammalian cells."

Reviewer #2 (Comments to the Authors (Required)):

I am satisfied with most of the authors' responses except for the answer to the second question of my previous comments. My original question intended to bring the authors' attention to the origin of the diameter difference of the microtubules which might lead to more structural and biological insight, but I did not anticipate the possible difference in protofilament numbers of the microtubules that the authors are now considering. I suggest the authors consider the following:

Major concern:

The following newly added text by the authors may be confusing to the readers: "The different microtubule diameters of the Class I and Class II filament containing structures suggest the possibility that there may be different microtubule protofilament numbers associated with the two classes. The measured Class I diameters are close to those of a 13 protofilament microtubule (Zhang et al., 2018) which may suggest that Class II filament containing microtubules have 14 protofilaments or an expansion in the 13 protofilament lattice".

We know microtubules in mammalian cells contains 13 protofilaments different from the microtubules polymerized from purified tubulin proteins. Have the authors seen any microtubules of 14 protofilaments in the cryo-tomograms of the treated HAP1 cells? It should not take much effort to examine the cryo-tomograms and find the answer. This can be done by looking into the cross-section view using the Slicer function of IMOD with a large thickness (similar to an end-on projection of short segment). If there are indeed 14-protofilament microtubules found in mammalian cells after a kinesore treatment, this result itself is interesting and should be included in the manuscript. If all the microtubules of distinctive protofilaments in tomograms contains 13 protofilaments, the diameter difference has nothing to do with the protofilament number of the microtubules.

Minor concern:

I am not sure if the author's answer implies that the diameter measurements of the empty segments of microtubules did not provide a relatively consistent value. If this is the case, it is

possible that the diameter of the empty microtubule segment is defined by the segments containing F-actins of the same microtubule. In addition, the author may also want to consider the possible contribution from microtubule lateral deformations induced by surrounding stress (see Amos, Structure, 2010; Sui and Downing, Structure, 2010), if they used longitudinal projections of the microtubule tomograms to measure the diameters. In fact, the presence of the F-actin filaments in the lumens of microtubules may increase the rigidity of microtubules by limiting their lateral deformation ability, which might be worth mentioning to strengthen the structural insight and biological implication of the manuscript. It is OK that the authors prefer not to include the diameter comparison with the empty microtubule segments. The comparison is useful, but is not an essential piece of information for the main points of this paper. There lacks details in the method section on how the outer and luminal diameters of the microtubules were measured.

Reviewer #3 (Comments to the Authors (Required)):

The revised manuscript by Paul et al. is significantly improved compared to the previous version. It took all my suggestions and fully addressed all the concerns I raised previously. I am convinced that there is actin filament inside microtubule. Therefore, I think this manuscript is now suitable for publication in JCB.